# Focused Quantization for Sparse CNNs

**Yiren Zhao**[*1]  **Xitong Gao**[*2]  **Daniel Bates**[1]  **Robert Mullins**[1]  **Cheng-Zhong Xu**[3]

[1] University of Cambridge
[2] Shenzhen Institutes of Advanced Technology
[3] University of Macau

## Abstract

Deep convolutional neural networks (CNNs) are powerful tools for a wide range of vision tasks, but the enormous amount of memory and compute resources required by CNNs pose a challenge in deploying them on constrained devices. Existing compression techniques, while excelling at reducing model sizes, struggle to be computationally friendly. In this paper, we attend to the statistical properties of sparse CNNs and present focused quantization, a novel quantization strategy based on power-of-two values, which exploits the weight distributions after fine-grained pruning. The proposed method dynamically discovers the most effective numerical representation for weights in layers with varying sparsities, significantly reducing model sizes. Multiplications in quantized CNNs are replaced with much cheaper bit-shift operations for efficient inference. Coupled with lossless encoding, we built a compression pipeline that provides CNNs with high compression ratios (CR), low computation cost and minimal loss in accuracy. In ResNet-50, we achieved a $18.08\times$ CR with only $0.24\%$ loss in top-5 accuracy, outperforming existing compression methods. We fully compressed a ResNet-18 and found that it is not only higher in CR and top-5 accuracy, but also more hardware efficient as it requires fewer logic gates to implement when compared to other state-of-the-art quantization methods assuming the same throughput.

## 1 Introduction

Despite deep convolutional neural networks (CNNs) demonstrating state-of-the-art performance in many computer vision tasks, their parameter-rich and compute-intensive nature substantially hinders the efficient use of them in bandwidth- and power-constrained devices. To this end, recent years have seen a surge of interest in minimizing the memory and compute costs of CNN inference.

Pruning algorithms compress CNNs by setting weights to zero, thus removing connections or neurons from the models. In particular, fine-grained pruning [16, 6] provides the best compression by removing connections at the finest granularity, *i.e.* individual weights. Quantization methods reduce the number of bits required to represent each value, and thus further provide memory, bandwidth and compute savings. *Shift quantization* of weights, which quantizes weight values in a model to powers-of-two or zero, *i.e.* $\{0, \pm 1, \pm 2, \pm 4, \dots\}$, is of particular of interest, as multiplications in convolutions become much-simpler bit-shift operations. The computational cost in hardware can thus be significantly reduced without a detrimental impact on the model's task accuracy [26]. Fine-grained pruning, however, is often in conflict with quantization, as pruning introduces various degrees of sparsities to different layers [25, 19]. Linear quantization methods (integers) have uniform quantization levels and non-linear quantizations (logarithmic, floating-point and shift) have fine levels

around zero but levels grow further apart as values get larger in magnitude. Both linear and nonlinear quantizations thus provide precision where it is not actually required in the case of a pruned CNN. It is observed that empirically, very few non-zero weights concentrate around zero in some layers that are sparsified with fine-grained pruning (see Figure 1c for an example). Shift quantization is highly desirable as it can be implemented efficiently, but it becomes a poor choice for certain layers in sparse models, as most near-zero quantization levels are under-utilized (Figure 1d).

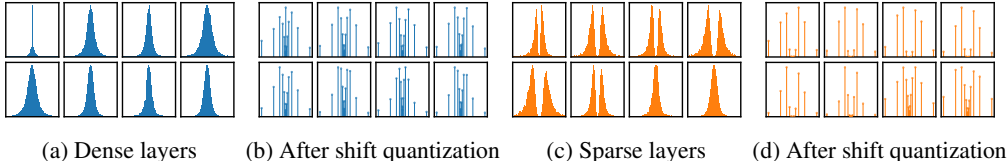

      (a) Dense layers      (b) After shift quantization     (c) Sparse layers     (d) After shift quantization

Figure 1: The weight distributions of the first 8 layers of ResNet-18 on ImageNet. (a) shows the weight distributions of the layers, (c) similarly shows the distributions (excluding zeros) for a sparsified variant. (b) and (d) respectively quantize the weight distributions on the left with 5-bit shift quantization. Note that in some sparse layers, greedy pruning encourages weights to avoid near zero values. Shift quantization on these layers thus results in poor utilization of the quantization levels.

This dichotomy prompts the question, *how can we quantize sparse weights efficiently and effectively?* Here, efficiency represents not only the reduced model size but also the minimized compute cost. Effectiveness means that the quantization levels are well-utilized. From an information theory perspective, it is desirable to design a quantization function $Q$ such that the quantized values in $\hat{\theta} = Q(\theta)$ closely match the prior weight distribution. We address both issues by proposing a new approach to quantize parameters in CNNs which we call *focused quantization* (FQ) that mixes *shift* and *recentralized* quantization methods. Recentralized quantization uses a mixture of Gaussian distributions to find the most concentrated probability masses in the weight distribution of sparse layers (first block in Figure 2), and independently quantizes the probability masses (rightmost of Figure 2) to powers-of-2 values. Additionally, not all layers consist of two probability masses, and recentralized quantization may not be necessary (as shown in Figure 1c). In such cases, we use the Wasserstein distance between the two Gaussian components to decide when to apply shift quantization.

For evaluation, we present a complete compression pipeline comprising fine-grain pruning, FQ and Huffman encoding and estimate the resource utilization in custom hardware required for inference. We show that the compressed models with FQ not only provide higher task accuracies, but also require less storage and lower logic usage when compared to other methods. This suggests the FQ-based compression is a more practical alternative design for future custom hardware accelerators designed for neural network inference [24].

In this paper, we make the following contributions:

- The proposed method, focused quantization for sparse CNNs, significantly reduces both computation and model size with minimal loss of accuracy.
- FQ is hybrid, it systematically mixes a recentralized quantization with shift quantization to provide the most effective quantization on sparse CNNs.
- We built a complete compression pipeline based on FQ. We observed that FQ achieves the highest compression rates on a range of modern CNNs with the least accuracy losses.
- We found that a hardware design based on FQ demonstrates the most efficient hardware utilization compared to previous state-of-the-art quantization methods [15, 23].

The rest of the paper is structured as follows. Section 2 discusses related work in the field of model compression. Section 3 introduces focused quantization. Section 4 presents and evaluates the proposed compression pipeline and Section 5 concludes the paper.

## 2 Related Work

Recently, a wide range of techniques have been proposed and proven effective for reducing the memory and computation requirements of CNNs. These proposed optimizations can provide direct

reductions in memory footprints, bandwidth requirements, total number of arithmetic operations, arithmetic complexities or a combination of these properties.

Pruning-based optimization methods directly reduce the number of parameters in a network. Fine-grained pruning method [6] significantly reduces the size of a model but introduces element-wise sparsity. Coarse-grained pruning [17, 4] shrinks model sizes and reduce computation at a higher granularity that is easier to accelerate on commodity hardware. Quantization methods allow parameters to be represented in more efficient data formats. Quantizing weights to powers-of-2 recently gained attention because it not only reduces the model size but also simplifies computation [14, 26, 18, 24]. Previous research also focused on quantizing CNNs to extremely low bit-widths such as ternary [27] or binary [12] values. They however introduce large numerical errors and thus cause significant degradations in model accuracies. To minimize loss in accuracies, the proposed methods of [23] and [15] quantize weights to $N$ binary values, compute $N$ binary convolutions and scale the convolution outputs individually before summation. Lossy and lossless encodings are other popular methods to reduce the size of a DNN, typically used in conjunction with pruning and quantization [3, 7].

Since many compression techniques are available and building a compression pipeline provides a multiplying effect in compression ratios, researchers start to chain multiple compression techniques. Han *et al.* [7] proposed Deep Compression that combines pruning, quantization and Huffman encoding. Dubey *et al.* [3] built a compression pipeline using their coreset-based filter representations. Tung *et al.* [22] and Polino *et al.* [20] integrated multiple compression techniques, where [22] combined pruning with quantization and [20] employed knowledge distillation on top of quantization. Although there are many attempts in building an efficient compression pipeline, the statistical relationship between pruning and quantization lacked exploration. In this paper, we look at exactly this problem and propose a new method that exploits the statistical properties of weights in pruned models to quantize them efficiently and effectively.

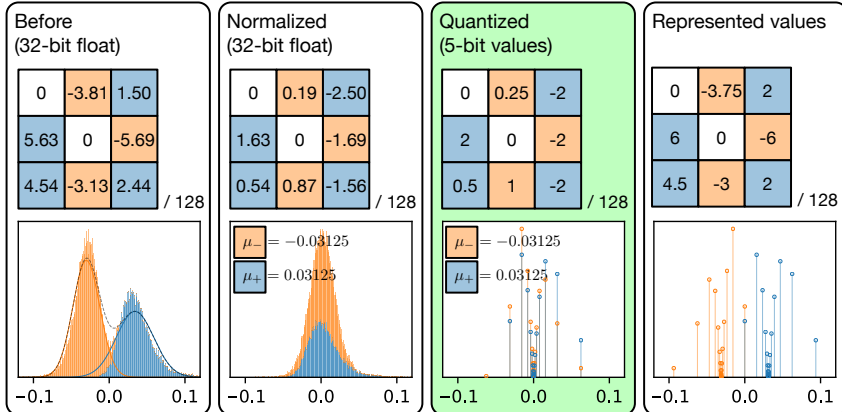

Figure 2: The step-by-step process of recentralized quantization of unpruned weights on `block3f/conv1` in sparse ResNet-50. Each step shows how it changes a filter and the distribution of weights. Higher peaks in the histograms denote values found with higher frequency. Values in the filter share a common denominator 128, indicated by "/128". The first estimates the high-probability regions with a Gaussian mixture, and assign weights to a Gaussian component. The second normalizes each weight. The third quantizes the normalized values with shift quantization and produces a representation of quantized weights used for inference. The final block visualizes the actual numerical values after quantization.

## 3 Method

### 3.1 Preliminaries: Shift quantization

Shift quantization is a quantization scheme which constrains weight values to powers-of-two or zero values. A representable value in a $(k+2)$-bit shift quantization is given by:

$$v = s \cdot 2^{e-b}, \tag{1}$$

where $s = \{-1, 0, 1\}$ denotes either zero or the sign of the value, $e$ is an integer bounded by $[0, 2^k - 1]$, and $b$ is the bias, a layer-wise constant which scales the magnitudes of quantized values. We use

$\hat{\theta} = Q_{n,b}^{\text{shift}}[\theta]$ to denote a n-bit shift quantization with a bias b of a weight value $\theta$ to the nearest representable value $\hat{\theta}$. As we have discussed earlier and illustrated in Figure 1, shift quantization on sparse layers makes poor use of the range of representable values, *i.e.* the resulting distribution after quantization $q_{n,b}^{\text{shift}}(\theta)$ is a poor approximation of the original layer weight distribution $p(\theta|\mathcal{D})$, where $\mathcal{D}$ is the training dataset.

## 3.2 Designing the Recentralized Quantization Function

Intuitively, it is desirable to concentrate quantization effort on the high probability regions in the weight distribution in sparse layers. By doing so, we can closely match the distribution of quantized weights with the original, and thus at the same time incur smaller round-off errors. Recentralized quantization $Q[\theta]$ is designed specifically for this purpose, and applied in a layer-wise fashion. Assuming that $\theta \in \boldsymbol{\theta}$ is a weight value of a convolutional layer, we can define $Q[\theta]$ as follows:

$$Q[\theta] = z_\theta \alpha \sum_{c \in C} \delta_{c,m_\theta} Q_c^{\text{rec}}[\theta], \text{ where } Q_c^{\text{rec}}[\theta] = Q_{n,b}^{\text{shift}} \left[ \frac{\theta - \mu_c}{\sigma_c} \right] \sigma_c + \mu_c. \tag{2}$$

Here $z_\theta$ is a predetermined constant $\{0, 1\}$ binary value to indicate if $\theta$ is pruned, and it is used to set pruned weights to 0. The set of components $c \in C$ determines the locations to focus quantization effort, each specified by the component's mean $\mu_c$ and standard deviation $\sigma_c$. The Kronecker delta $\delta_{c,m_\theta}$ evaluates to either 1 when $c = m_\theta$, or 0 otherwise. In other words, the constant $m_\theta \in C$ chooses which component in $C$ is used to quantize $\theta$. Finally, $Q_c^{\text{rec}}[\theta]$ locally quantizes the component $c$ with shift quantization. Following [27] and [14], we additionally introduce a layer-wise learnable scaling factor $\alpha$ initialized to 1, which empirically improves the task accuracy.

By adjusting the $\mu_c$ and $\sigma_c$ of each component $c$, and finding suitable assignments of weights to the components, the quantized weight distribution $q_\phi(\theta)$ can thus match the original closely, where we use $\phi$ as a shorthand to denote the relevant hyperparameters, *e.g.* $\mu_c, \sigma_c$. The following section explains how we can optimize them efficiently.

## 3.3 Optimizing Recentralized Quantization $Q[\theta]$

Hyperparameters $\mu_c$ and $\sigma_c$ in recentralized quantization can be optimized by applying the following two-step process in a layer-wise manner, which first identifies regions with high probabilities (first block in Figure 2), then locally quantize them with shift quantization (second and third blocks in Figure 2). First, we notice that in general, the weight distribution resembles a mixture of Gaussian distributions. It is thus more efficient to find a Gaussian mixture model $q_\phi^{\text{mix}}(\theta)$ that approximates the original distribution $p(\theta|\mathcal{D})$ to closely optimize the above objective:

$$q_\phi^{\text{mix}}(\theta) = \sum_{c \in C} \lambda_c f(\theta|\mu_c, \sigma_c), \tag{3}$$

where $f(\theta|\mu_c, \sigma_c)$ is the probability density function of the Gaussian distribution $\mathcal{N}(\mu_c, \sigma_c)$, the non-negative $\lambda_c$ defines the mixing weight of the $c^{\text{th}}$ component and $\Sigma_{c \in C} \lambda_c = 1$. Here, we find the set of hyperparameters $\mu_c, \sigma_c$ and $\lambda_c$ contained in $\phi$ that maximizes $q_\phi^{\text{mix}}(\theta)$ given that $\theta \sim p(\theta|\mathcal{D})$. This is known as the *maximum likelihood estimate* (MLE), and the MLE can be efficiently computed by the *expectation-maximization* (EM) algorithm [1]. In practice, we found it sufficient to use two Gaussian components, $C = \{-, +\}$, for identifying high-probability regions in the weight distribution. For faster EM convergence, we initialize $\mu_-, \sigma_-$ and $\mu_+, \sigma_+$ respectively with the means and standard deviations of negative and positive values in the layer weights respectively, and $\lambda_-, \lambda_+$ with $\frac{1}{2}$.

We then generate $m_\theta$ from the mixture model, which individually selects the component to use for each weight. For this, $m_\theta$ is evaluated for each $\theta$ by sampling a categorical distribution where the probability of assigning a component $c$ to $m_\theta$, *i.e.* $p(m_\theta = c)$, is $\lambda_c f(\theta|\mu_c, \sigma_c) / q_\phi^{\text{mix}}(\theta)$.

Finally, we set the constant b to a powers-of-two value, chosen to ensure that $q_{n,b}^{\text{shift}}[\cdot]$ allows at most a proportion of $\frac{1}{2^n+1}$ values to overflow and clips them to the maximum representable magnitude. In practice, this heuristic choice makes better use of the quantization levels provided by shift quantization than disallowing overflows. After determining all of the relevant hyperparameters with the method described above, $\hat{\theta} = Q[\theta]$ can be evaluated to quantize the layer weights.

### 3.4 Choosing the Appropriate Quantization

As we have discussed earlier, the weight distribution of sparse layers may not always have multiple high-probability regions. For example, fitting a mixture model of two Gaussian components on the layer in Figure 3a gives highly overlapped components. It is therefore of little consequence which component we use to quantize a particular weight value. Under this scenario, we can simply use n-bit shift quantization $Q_{n,b}^{shift}[\cdot]$ instead of a n-bit $Q[\cdot]$ which internally uses a $(n-1)$-bit signed shift quantization. By moving the 1 bit used to represent the now absent m to shift quantization, we further increase its precision.

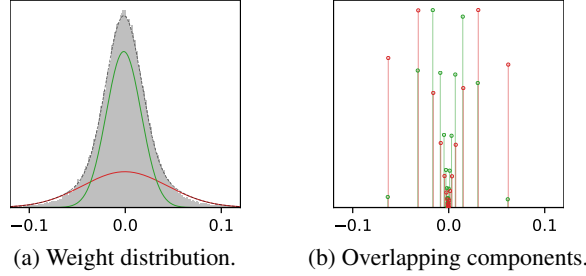

(a) Weight distribution.  (b) Overlapping components.

Figure 3: The weight distribution of the layer `block22/conv1` in a sparse ResNet-18 trained on ImageNet, as shown by the histograms. It shows that when the two Gaussian components have a large overlap, quantizing with either one of them results in almost the same quantization levels.

To decide whether to use shift or recentralized quantization, it is necessary to introduce a metric to compare the similarity between the pair of components. While the KL-divergence provides a measure for similarity, it is however non-symmetric, making it unsuitable for this purpose. To address this, we propose to first normalize the distribution of the mixture, then to use the 2-Wasserstein metric between the two Gaussian components after normalization as a decision criterion, which we call the *Wasserstein separation*:

$$\mathcal{W}(c_1, c_2) = \frac{1}{\sigma^2}\left( (\mu_{c_1} - \mu_{c_2})^2 + (\sigma_{c_1} - \sigma_{c_2})^2 \right), \tag{4}$$

where $\mu_c$ and $\sigma_c$ are respectively the mean and standard deviation of the component $c \in \{c_1, c_2\}$, and $\sigma^2$ denotes the variance of the entire weight distribution. FQ can then adaptively pick to use recentralized quantization for all sparse layers except when $\mathcal{W}(c_1, c_2) < w_{sep}$, and shift quantization is used instead. In our experiments, we found $w_{sep} = 2.0$ usually provides a good decision criterion. In Section 4.3, we additionally study the impact of quantizing a model with different $w_{sep}$ values.

### 3.5 Model Optimization

To optimize the quantized sparse model, we integrate the quantization process described above into the gradient-based training of model parameters. Initially, we compute the hyperparameters $\mu_c, \sigma_c, \lambda_c$ for each layer, and generate the component selection mask $m_\theta$ for each weight $\theta$ with the method in Section 3.3. The resulting model is then fine-tuned where the forward pass uses quantized weights $\hat{\theta} = Q[\theta]$, and the backward pass updates the floating-point weight parameters $\theta$ by treating the quantization as an identity function. During the fine-tuning process, the hyperparameters used by $Q[\theta]$ are updated using the current weight distribution at every $k$ epochs. We also found that in our experiments, exponentially increasing the interval $k$ between consecutive hyperparameter updates helps to reduce the variance introduced by sampling and improves training quality.

### 3.6 The MDL Perspective

Theoretically, the model optimization can be formulated as a *minimum description length* (MDL) optimization [10, 5]. Given that we approximate the posterior $p(\theta|\mathcal{D})$ with a distribution of quantized weights $q_\phi(\theta)$, where $\phi$ contains the hyperparameters used by the quantization function $Q[\theta]$, the MDL problem minimizes the *variational free energy* [5], $\mathcal{L}(\boldsymbol{\theta}, \boldsymbol{\alpha}, \boldsymbol{\phi}) = \mathcal{L}_E + \mathcal{L}_C$, where:

$$\mathcal{L}_E = \mathbb{E}_{\hat{\boldsymbol{\theta}} \sim q_\phi(\theta)}\left[ -\log p(\mathbf{y}|\mathbf{x}, \boldsymbol{\alpha}, \hat{\boldsymbol{\theta}}) \right], \quad \mathcal{L}_C = \mathrm{KL}\left( q_\phi(\theta) \| p(\theta|\mathcal{D}) \right). \tag{5}$$

The error cost $\mathcal{L}_{\mathrm{E}}$ reflects the cross-entropy loss of the quantized model, with quantized weights $\hat{\theta}$ and layer-wise scalings $\boldsymbol{\alpha}$, trained on the dataset $\mathcal{D} = (\mathbf{x}, \mathbf{y})$, which is optimized by stochastic gradient descent. The complexity cost $\mathcal{L}_{\mathrm{C}}$ is the *Kullback-Leibler* (KL) divergence from the quantized weight distribution to the original. Intuitively, minimizing $\mathcal{L}_{\mathrm{C}}$ reduces the discrepancies between the weight distributions before and after quantization. As this is intractable, we replace $q_{\boldsymbol{\phi}}(\theta)$ with a close surrogate, a Gaussian mixture $q_{\boldsymbol{\phi}}^{\mathrm{mix}}(\theta)$. It turns out that the process of finding the MLE discussed in Section 3.3 is equivalent to minimizing $\mathrm{KL}\left(q_{\boldsymbol{\phi}}^{\mathrm{mix}}(\theta)\|p(\theta|\mathcal{D})\right)$, a close proxy for $\mathcal{L}_{\mathrm{C}}$. Section 3.5 then interleaves the optimization of $\mathcal{L}_{\mathrm{E}}$ and $\mathcal{L}_{\mathrm{C}}$ to minimize the MDL objective $\mathcal{L}(\boldsymbol{\theta}, \boldsymbol{\alpha}, \boldsymbol{\phi})$.

## 4 Evaluation

We applied *focused compression* (FC), a compression flow which consists of pruning, FQ and Huffman encoding, on a wide range of popular vision models including MobileNets [11, 21] and ResNets [8, 9] on the ImageNet dataset [2]. For all of these models, FC produced models with high compression ratios (CRs) and permitted a multiplication-free hardware implementation of convolution while having minimal impact on the task accuracy. In our experiments, models are initially sparsified using Dynamic Network Surgery [6]. FQ is subsequently applied to restrict weights to low-precision values. During fine-tuning, we additionally employed Incremental Network Quantization (INQ) [26] and gradually increased the proportion of weights being quantized to 25%, 50%, 75%, 87.5% and 100%. At each step, the models were fine-tuned for 3 epochs at a learning rate of 0.001, except for the final step at 100% we ran for 10 epochs, and decay the learning rate every 3 epochs. Finally, Huffman encoding was applied to model weights which further reduced model sizes. To simplify inference computation in custom hardware (Section 4.2), in our experiments $\mu_-$ and $\mu_+$ are quantized to the nearest powers-of-two values, and $\sigma_-$ and $\sigma_+$ are constrained to be equal.

### 4.1 Model Size Reduction

Table 1 compares the accuracies and compression rates before and after applying the compression pipeline under different quantization bit-widths. It demonstrates the effectiveness of FC on the models. We found that sparsified ResNets with 7-bit weights are at least $16\times$ smaller than the original dense model with marginal degradations ($\leq 0.24\%$) in top-5 accuracies. MobileNets, which are much less redundant and more compute-efficient models to begin with, achieved a smaller CR at around $8\times$ and slightly larger accuracy degradations ($\leq 0.89\%$). Yet when compared to the ResNet-18 models, it is not only more accurate, but also has a significantly smaller memory footprint at 1.71 MB.

In Table 2 we compare FC with many state-of-the-art model compression schemes. It shows that FC simultaneously achieves the best accuracies and the highest CR on both ResNets. Trained Ternary Quantization (TTQ) [27] quantizes weights to ternary values, while INQ [26] and extremely low bit neural network (denoted as ADMM) [14] quantize weights to ternary or powers-of-two values using shift quantization. Distillation and Quantization (D&Q) [20] quantize parameters to integers via distillation. Note that D&Q's results used a larger model as baseline, hence the compressed model has high accuracies and low CR. We also compared against Coreset-Based Compression [3] comprising pruning, filter approximation, quantization and Huffman encoding. For ResNet-50, we additionally compare against ThiNet [17], a filter pruning method, and Clip-Q [22], which interleaves training steps with pruning, weight sharing and quantization. FC again achieves the highest CR ($18.08\times$) and accuracy ($74.86\%$).

### 4.2 Computation Reduction

Quantizing weights using FC can significantly reduce computation complexities in models. By further quantizing activations and BN parameters to integers, the expensive floating-point multiplications and additions in convolutions can be replaced with simple bit-shift operations and integer additions. This can be realized with much faster software or hardware implementations, which directly translates to energy saving and much lower latencies in low-power devices. In Table 3, we evaluate the impact on accuracies by progressively applying FQ on weights, and integer quantizations on activations and batch normalization (BN) parameters. It is notable that the final fully quantized model achieve similar accuracies to LQ-Net.

Table 1: The accuracies (%), sparsities (%) and CRs of focused compression on ImageNet models. The baseline models are dense models before compression and use 32-bit floating-point weights, and 5 bits and 7 bits denote the number of bits used by individual weights of the quantized models before Huffman encoding.

| Model | Top-1 | $\Delta$ | Top-5 | $\Delta$ | Sparsity | Size (MB) | CR ($\times$) |
|---|---|---|---|---|---|---|---|
| ResNet-18 | 68.94 | — | 88.67 | — | 0.00 | 46.76 | — |
| Pruned | 69.24 | 0.30 | 89.05 | 0.38 | 74.86 | 8.31 | 5.69 |
| 5 bits | 68.36 | -0.58 | 88.45 | -0.22 | 74.86 | 2.86 | 16.33 |
| 7 bits | 68.57 | -0.37 | 88.53 | -0.14 | 74.86 | 2.94 | 15.92 |
| ResNet-50 | 75.58 | — | 92.83 | — | 0.00 | 93.82 | — |
| Pruned | 75.10 | -0.48 | 92.58 | -0.25 | 82.70 | 11.76 | 7.98 |
| 5 bits | 74.86 | -0.72 | 92.59 | -0.24 | 82.70 | 5.19 | 18.08 |
| 7 bits | 74.99 | -0.59 | 92.59 | -0.24 | 82.70 | 5.22 | 17.98 |
| MobileNet-V1 | 70.77 | — | 89.48 | — | 0.00 | 16.84 | — |
| Pruned | 70.03 | -0.74 | 89.13 | -0.35 | 33.80 | 6.89 | 2.44 |
| 7 bits | 69.13 | -1.64 | 88.61 | -0.87 | 33.80 | 2.13 | 7.90 |
| MobileNet-V2 | 71.65 | — | 90.44 | — | 0.00 | 13.88 | — |
| Pruned | 71.24 | -0.41 | 90.31 | -0.13 | 31.74 | 5.64 | 2.46 |
| 7 bits | 70.05 | -1.60 | 89.55 | -0.89 | 31.74 | 1.71 | 8.14 |

Table 2: Comparisons of top-1 and top-5 accuracies (%) and CRs with various compression methods. Numbers with $\star$ indicate results not originally reported and calculated by us. Note that D&Q used a much larger ResNet-18, the 5 bases used by ABC-Net denote 5 separate binary convolutions. LQ-Net used a "pre-activation" ResNet-18 [9] with a 1.4% higher accuracy baseline than ours.

| ResNet-18 | Top-1 | Top-5 | Size (MB) | CR ($\times$) |
|---|---|---|---|---|
| TTQ [27] | 66.00 | 87.10 | 2.92$\star$ | 16.00$\star$ |
| INQ (2 bits) [26] | 66.60 | 87.20 | 2.92$\star$ | 16.00$\star$ |
| INQ (3 bits) [26] | 68.08 | 88.36 | 4.38$\star$ | 10.67$\star$ |
| ADMM (2 bits) [14] | 67.0 | 87.5 | 2.92$\star$ | 16.00$\star$ |
| ADMM (3 bits) [14] | 68.0 | 88.3 | 4.38$\star$ | 10.67$\star$ |
| ABC-Net (5 bases, or 5 bits) [15] | 67.30 | 87.90 | 7.30$\star$ | 6.4 $\star$ |
| LQ-Net (preact, 2 bits) [23] | 68.00 | 88.00 | 2.92$\star$ | 16.00$\star$ |
| D&Q (large) [20] | **73.10** | **91.17** | 21.98 | 2.13$\star$ |
| Coreset [3] | 68.00 | — | 3.11$\star$ | 15.00 |
| Focused compression (5 bits, sparse) | 68.36 | 88.45 | **2.86** | **16.33** |
| **ResNet-50** | **Top-1** | **Top-5** | **Size (MB)** | **CR ($\times$)** |
| INQ (5 bits) [26] | 74.81 | 92.45 | 14.64$\star$ | 6.40$\star$ |
| ADMM (3 bits) [14] | 74.0 | 91.6 | 8.78$\star$ | 10.67$\star$ |
| ThiNet [17] | 72.04 | 90.67 | 16.94 | 5.53$\star$ |
| Clip-Q [22] | 73.70 | — | 6.70 | 14.00$\star$ |
| Coreset [3] | 74.00 | — | 5.93$\star$ | 15.80 |
| Focused compression (5 bits, sparse) | **74.86** | **92.59** | **5.19** | **18.08** |

Figure 4 shows an accelerator design of the dot-products used in the convolutional layers with recentralized quantization for inference. Using this, in Table 4 we provide the logic usage required by the implementation to compute a convolution layer with $3 \times 3$ filters with a padding size of 1, which takes as input a $8 \times 8 \times 100$ activation and produce a $8 \times 8 \times 100$ tensor output. Additionally, we compare FQ to shift quantization, ABC-Net [15] and LQ-Net [23]. The #Gates indicates the lower bound on the number of two-input logic gates required to implement the custom hardware accelerators for the convolution, assuming an unrolled architecture and the same throughput. Internally, a 5-bit FQ-based inference uses 3-bit unsigned shift quantized weights, with a minimal overhead for the added logic. Scaling constants $\sigma_-$ and $\sigma_+$ are equal and thus can be fused into $\alpha_l$. Perhaps most surprisingly, a 5-bit FQ has more quantization levels yet uses fewer logic gates, when compared to ABC-Net and LQ-Net implementing the same convolution but with different quantizations. Both ABC-Net and LQ-Net quantize each weight to $N$ binary values, and compute $N$ parallel binary convolutions for each binary weight. The $N$ outputs are then accumulated for each pixel in the output feature map. In Table 4, they use $N = 5$ and 2 respectively. Even with the optimal compute pattern proposed by the two methods, there are at least $O(MN)$ additional high-precision multiply-adds, where $M$ is the number of parallel binary convolutions, and $N$ is the number of output pixels. This overhead is much more significant when compared to other parts of compute in the convolution. As shown in Table 4, both have higher logic usage because of the substantial amount of high-precision multiply-adds. In contrast, FQ has only one final learnable layer-wise scaling multiplication that can be further optimized out as it can be fused into BN for inference. Despite having more quantization levels and a higher CR, and being more efficient in hardware resources, the fully quantized ResNet-18 in Table 3 can still match the accuracy of a LQ-Net ResNet-18.[2]

Table 3: Comparison of the original ResNet-18 with successive quantizations applied on weights, activations and BN parameters. Each row denotes added quantization on new components.

| Quantized | Top-1 | $\Delta$ | Top-5 | $\Delta$ |
|---|---|---|---|---|
| Baseline | 68.94 | — | 88.67 | — |
| + Weights (5-bit FQ) | 68.36 | -0.58 | 88.45 | -0.22 |
| + Activations (8-bit integer) | 67.89 | -1.05 | 88.08 | -0.59 |
| + BN (16-bit integer) | 67.95 | -0.99 | 88.06 | -0.61 |

Table 4: Computation resource estimates of custom accelerators for inference assuming the same compute throughput.

| Configuration | #Gates | Ratio |
|---|---|---|
| ABC-Net (5 bases, or 5 bits) | 806.1 M | 2.93× |
| LQ-Net (2 bits) | 314.4 M | 1.14× |
| Shift quantization (3 bits, unsigned) | 275.2 M | 1.00× |
| FQ (5 bits) | 275.6 M | 1.00× |
| FQ (5 bits) + Huffman | 276.4 M | 1.00× |

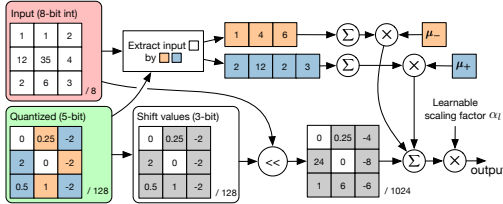

Figure 4: An implementation of the dot-product used in convolution between an integer input and a filter quantized by recentralized quantization. The notation $/N$ means the filter values share a common denominator $N$.

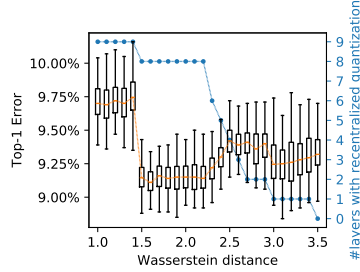

Figure 5: The effect of different threshold values on the Wasserstein distance. The larger the threshold, the fewer the number of layers using recentralized quantization instead of shift quantization.

## 4.3 Exploring the Wasserstein Separation

In Section 3.4, we mentioned that some of the layers in a sparse model may not have multiple high-probability regions. For this reason, we use the Wasserstein distance $\mathcal{W}(c_1, c_2)$ between the two components in the Gaussian mixture model as a metric to differentiate whether recentralized or shift quantization should be used. In our experiments, we specified a threshold $w_{\text{sep}} = 2.0$ such that for each layer, if $\mathcal{W}(c_1, c_2) \geq w_{\text{sep}}$ then recentralized quantization is used, otherwise shift quantization is employed instead. Figure 5 shows the impact of choosing different $w_{\text{sep}}$ ranging from 1.0 to 3.5 at 0.1 increments on the Top-1 accuracy. This model is a fast CIFAR-10 [13] classifier with only 9 convolutional layers, so that it is possible to repeat training 100 times for each $w_{\text{sep}}$ value to produce high-confidence results. Note that the average validation accuracy is minimized when the layer with only one high-probability region uses shift quantization and the remaining 8 use recentralized quantization, which verifies our intuition.

## 5 Conclusion

In this paper, we exploit the statistical properties of sparse CNNs and propose focused quantization to efficiently and effectively quantize model weights. The quantization strategy uses Gaussian mixture models to locate high-probability regions in the weight distributions and quantize them in fine levels. Coupled with pruning and encoding, we build a complete compression pipeline and demonstrate high compression ratios on a range of CNNs. In ResNet-18, we achieve $18.08\times$ CR with minimal loss in accuracies. We additionally show FQ allows a design that is more efficient in hardware resources. Furthermore, the proposed quantization uses only powers-of-2 values and thus provides an efficient compute pattern. The significant reductions in model sizes and compute complexities can translate to direct savings in power efficiencies for future CNN accelerators on IoT devices. Finally, FQ and the optimized models are fully open-source and released to the public[3].

## Acknowledgments

This work is supported in part by the National Key R&D Program of China (No. 2018YFB1004804), the National Natural Science Foundation of China (No. 61806192). We thank EPSRC for providing Yiren Zhao his doctoral scholarship.

## Footnotes

*Xitong Gao and Yiren Zhao contributed equally to this work. Correspondence to Xitong Gao (xt.gao@siat.ac.cn) and Yiren Zhao (yiren.zhao@cl.cam.ac.uk).

[2]It is also notable that LQ-Net used "pre-activation" ResNet-18 which has a 1.4% advantage in baseline accuracy compared to ours.

[3]Available at: https://github.com/deep-fry/mayo.

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
