[Supplementary Material · Supplementary.pdf]

# Focused Quantization for Sparse DNNs
# Supplementary Material

## 1 Training Configuration

For image preprocessing, we follow the augmentation procedures in Krizhevsky *et al.* [2012], which includes aspect ratio distortion, random flipping, random cropping, and hue, saturation, contrast and brightness changes to preprocess each training example.

## 2 Model Optimization

Here we provide the details of the model optimization explained in Section 3.5 in the form of an algorithm. Algorithm 1 optimizes $\mathcal{L}(\boldsymbol{\theta}, \boldsymbol{\phi})$, where $E$ specifies the number of epochs to fine-tune the quantized sparse model, and it returns the final optimized hyperparameters $\boldsymbol{\phi}^\star$ and quantized weights $\mathbf{Q}_{\boldsymbol{\phi}^\star}[\boldsymbol{\theta}]$. Note that we assume the pruned weights given by the pruning constant $z_\theta$ to remain zero throughout fine-tuning.

---

**Algorithm 1** Model Optimization

---

1: **function** OPTIMIZE($\boldsymbol{\theta}, E$)
2:      $e \leftarrow 0, k \leftarrow 1$
3:      **while** $e < E$ **do**
4:          $\boldsymbol{\phi}^\star \leftarrow \mathrm{argmin}_{\boldsymbol{\phi}} \, \mathrm{KL}\left(q_{\boldsymbol{\phi}}^{\mathrm{mix}}(\theta) \| p(\theta)\right)$
5:          **for** $\theta \in \boldsymbol{\theta}$ **do**
6:              Sample the component selector $\mathrm{m}_\theta$ in $\boldsymbol{\phi}^\star$
7:          **end for**
8:          **for** $k$ epochs **do**
9:              Sample a mini-batch $(\tilde{\mathbf{x}}, \tilde{\mathbf{y}})$ from $\mathcal{D}$
10:              $\boldsymbol{\theta} \leftarrow \mathrm{SGD}\left(-\log p\left(\tilde{\mathbf{y}} | \tilde{\mathbf{x}}, \mathbf{Q}_{\boldsymbol{\phi}^\star}[\boldsymbol{\theta}]\right)\right)$
11:          **end for**
12:          $e \leftarrow e + k, k \leftarrow 2k$
13:      **end while**
14:      **return** $\boldsymbol{\phi}^\star, \mathbf{Q}_{\boldsymbol{\phi}^\star}[\boldsymbol{\theta}]$
15: **end function**

---

For ResNet-50 on ImageNet, line 4 in the algorithm above takes 24 minutes to complete on an Intel Core i7-6700k CPU, while each epoch of the SGD optimization (line 8–11) requires 1.5 GPU-day to complete on an Nvidia GTX 1080 Ti. For each Image model we fine-tune for 10 epochs.

## 3   Bit-width Saving Tricks

Recentralized quantization Q is designed to capture the high-probability components in the weight distribution, which in theory provides a less redundant use of bits compared to shift quantization. We further reduce the bit-width by removing certain representable values that occur rarely after quantization. Although it does not bring better compression rates for Huffman-coded weights because we are removing rarely used values, it lowers the number of bits required for representing weights assuming constant bit-widths.

The tricks are generally applicable. Consider the $c_-$ (orange) and $c_+$ (blue) Gaussian components in the first block of Figure 2 in the paper, it is notable that the means $\mu_-$ and $\mu_+$ are surrounded with many fine-grained quantization levels, thus sacrificing these representations by quantizing to nearby values is equivalently efficient. Similarly, very few values quantized by $c_-$ lie about the well-quantized region of $c_+$ and *vice versa*. It means that we can remove the largest representation from $c_-$ and smallest representation from $c_+$. By removing these values from the representation, we use exactly *at most* n bits to represent a Q quantized value which internally uses $(n-1)$-bit shift quantization. To further simplify computation, we constrain $\sigma_-$ and $\sigma_+$ to the nearest powers-of-two values. For instance, a 3-bit recentralized quantization uses the following representable values $\{-9, -5, 3\} \cup \{-3, 5, 9\} \cup \{0\}$ if $\alpha_l = 1, \mu_- = -1, \mu_+ = 1, b = 0$, where the first two sets correspond to values quantized by the $c_-$ and $c_+$ components respectively.

## References

Alex Krizhevsky, Ilya Sutskever, and Geoffrey E Hinton. Imagenet classification with deep convolutional neural networks. In *Advances in Neural Information Processing Systems 25*. 2012.