[Reviews · NeurIPS 2019]

Reviewer 1



Summary of the paper: This paper proposes a distribution aware quantization which chooses between recentralized and shift quantizations based on weight distributions in the kernels. The end-to-end compression pipeline, when used on multiple architectures, shows good compression ratios over the original models. Finally, the new implementation proposed utilizes the maximum hardware optimization than the earlier proposed methods. Originality: The ideas in general based on distributions of weights have been around but the proposed methods is novel in the way it uses the weight distributions in the sparse CNNs to get optimal quantization strategies. This work is novel and grounded to the best of my knowledge. Quality: The experimental results are good but can be improved by the suggestion following in the review. The idea is simple and easy to follow. I think it is a good quality of work which will help the community. Clarity and writing: The paper was easy to follow and understand barring a few minor things: 1) The citation style is different from the normal style of neurips for eg. [1], [3,4,5]. Instead, the authors have used abc et al., format which takes up space. Please consider moving to the other format. 2) In Figure 2, I am not sure what / 128 means. It would be good to explain what that means for the ease of the reader. 3) The gap between table1 and its caption and table 2 and caption are very low. Please reformat it to make it look clean. 4) Please change the name of the paper from "Focused Quantization for Sparse DNNs" to "Focused Quantization for Sparse CNNs" in the manuscript. I think that is a mistake overlooked while submission. Positives: 1) Good idea and nice utilization of weight distributions of sparse CNNs to get the right quantization technique based on bit shifts. 2) Extensive experimental results and good ablation study. Issues and questions: 1) The only major issue according to me is that not all compression techniques provide good computation gains as well due to a lot of constraints. In order to make any of the claims made in the paper about efficiency, it would be good to have inference time for each of the models reported in table 1 and table 2. - this will make the experiments stronger 2) In conclusion, there is a claim about savings in power efficiencies for future CNN accelerators in IoT devices. Given IoT devices mostly are single-threaded it would be good if the authors could provide an example of what they are talking (expecting in the future) about and to make it even stronger a real-world deployment and case study will make the papers results stand out. ------------------------------------------------------------------------------------------------------------------------------------------------------------------------------------------------------------------------- I have read the author response thoroughly and I am happy with the response. The response addressed my concerns and I would like to see the #gates for the 8-bit quantized W+A in Table #4 in the camera-ready. I also agree with the other reviewers' comments and would like to see them addressed completely as done in rebuttal in the camera-ready.

Reviewer 2



Originality: I believe the quantization approach proposed in this paper is novel, and provides a new general framework to quantize sparse CNNs. Quality: The overall quality of this paper is good. The experimental results are detailed and provide solid support to the effectiveness of the proposed approach. However, I think the experimental section can be further strengthen by considering more datasets. Clarity: The paper is very well-written. In particular, it shows the intuition behind the proposed method, and how the proposed method relates to existing machine learning frameworks such as minimum description length optimization. Significance: I believe the paper shows promising results on addressing the problem of quantizing sparse CNNs.

Reviewer 3



This work treats the sparse weight distribution as a mixture of Gaussian distribution, and use Focused Quantization to quantize the sparse weights. The integrated quantization and pruning method help compress the model size. Recentralized Quantization uses MLE to find the appropriate hyperparameters and Shift Quantization uses powers-of-2 values to quantize a single high-probability region. In terms of the compression algorithm, the idea is novel and makes a lot of sense. In Section 3.6, there is a small mistake, the variational free energy should be represented as L (θ, α, Φ). Since α is responsible for the cross-entropy loss and Φ is composed of hyperparameters that are related to the discrepancies between the weight distributions before and after quantization, it is not included in the set Φ. In terms of the computation reduction, the powers-of-2 values can replace integer multiplications with shift operations which is cheap for hardware implementation. However, in this work, the whole pipeline is too complicated. In line 258, the authors mentioned that they have only one high-precision computing unit. However, the Recentralized Quantization actually introduces two high-precision parameters, μ, and σ (in addition, for each weight, these two params are different). The dot products in this paper, therefore, requires more high-precision multiply-adds units. This paper is well-written and easy to follow, except some figures are confusing. For example, in the last two blocks of Fig. 2, the meaning of y_axis should be defined clearly. Also in Fig.4, it seems that the standard deviation σ is missing. The experimental results are solid. However, this paper mainly presents a quantization method over sparse weights. The real effectiveness of this quantization scheme is not well evaluated. The authors should provide a comparison of their model between the model after pruning but before quantization to demonstrate the quantization performance.

[Author Response · NeurIPS 2019]

We thank the reviewers for the positive comments, and would like to provide answers to the questions raised.

We would like to first address Reviewer #1's questions regarding the inference time and a case study for IoT devices.
The reviewer states that not all compression techniques provide good computation gains due to a lot of constraints.
This is why in our evaluation, the target is a custom hardware implementation, and thus the issues above of efficiently
targeting a particular CPU or GPU don't apply on custom hardware. Instead, the #Gates estimations measure hardware
efficiency and they are derived from a hardware accelerator design mapped to a Stratix 10 FPGA device. We additionally
ensure that all designs with different quantization methods have the same processing performance (line 246 in the paper).
A hardware design using focused quantization (FQ) requires fewer gates than its counterparts (LQ-Net, ABC-Net) to
achieve the same performance, hence is more power efficient. Conversely, assuming that they all use the same amount
of logic resources, FQ thus has a higher throughput than the compared methods, as we can perform more computations
in parallel. While it is possible to provide certain performance statistics (*e.g.* energy consumed per image, frames per
second given the same chip area, *etc.*) for a particular ASIC design, yet the intricacy involved (*e.g.* the technology node,
circuit design, memory bandwidth, *etc.*) is beyond the scope of this paper. The reviewer suggests us to provide an
example of how our technique relates to future deployment of CNN accelerators in IoT devices. We believe future IoT
systems bring heterogeneous compute: CNN accelerators as co-processors to assist general purpose CPUs. Note that
IoT CNN accelerators today typically operate on common quantizations with higher bit-widths, *e.g.* 8-bit fixed-point
weights and activations in Eyeriss v2 [1]. The updated paper will address this and include in Table 4 additional #Gates
comparison with 8-bit fixed-point weights and activations.

Reviewer #3 suggested that as focused quantization introduces high-precision hyperparameters $\mu_-$, $\mu_+$, $\sigma_-$ and $\sigma_+$,
the dot products would require more high-precision multiply-adds. We would like to thank the reviewer for catching
this, and will clarify this accordingly. In all of our experiments in the supplementary source code, to simplify inference
computation, the hyperparameters $\mu_-$ and $\mu_+$ are quantized to the nearest powers-of-two values, and $\sigma_-$ and $\sigma_+$
are constrained to be equal in the optimization process. Effectively in our implementation of the dot-product, all
high-precision parameters, *i.e.* $\sigma_-$, $\sigma_+$ and the layer-wise learnable scalar $\alpha$, can thus be fused into batch normalization
during inference. The reviewer also pointed out that Figure 4 is missing $\sigma$ values, which can be explained by the same
reason above. We will clarify this in both Figure 2 and Figure 4.

In Table 1 below, we present the top-1 and top-5 accuracies for the pruned, encoded but not quantized models, as kindly
suggested by Reviewer #3. These results will later be added to the paper. We are working on 3-bit results as kindly
requested by Reviewer #2, and will include them in later changes.

Finally, we would like to additionally make the following minor changes to further improve the quality of the paper:

1. The notation of the form "/$N$" (*e.g.* "/128") in Figure 2 and Figure 4 denotes all numbers in the same
grid share a common denominator $N$, and thus the true values are the numbers scaled by $\frac{1}{N}$. We will add
descriptions to the figures to make it clear to the readers.

2. We will add the y-axis labels ("frequency") to Figure 2 and Figure 3 as kindly suggested by Reviewer #3.

3. We will adopt the numerical citation style, replace "DNNs" in the title with "CNNs", and fix caption spacing
and the notation as kindly suggested by Reviewer #1.

We thank the reviewers again for your detailed comments and help on improving the quality of our paper.

Table 1: Additional results for Table 1, newly added rows are shaded.

| Model | Top-1 | $\Delta$ | Top-5 | $\Delta$ | Sparsity (%) | Size (MB) | CR ($\times$) |
|---|---|---|---|---|---|---|---|
| ResNet-18 | 68.94 | — | 88.67 | — | 0.00 | 46.76 | — |
| Pruned | 69.24 | 0.30 | 89.05 | 0.38 | 74.86 | 8.31 | 5.69 |
| ResNet-50 | 75.58 | — | 92.83 | — | 0.00 | 93.82 | — |
| Pruned | 75.10 | -0.48 | 92.58 | -0.25 | 82.70 | 11.76 | 7.98 |
| MobileNet-V1 | 70.77 | — | 89.48 | — | 0.00 | 16.84 | — |
| Pruned | 70.03 | -0.74 | 89.13 | -0.35 | 33.80 | 6.89 | 2.44 |
| MobileNet-V2 | 71.65 | — | 90.44 | — | 0.00 | 13.88 | — |
| Pruned | 71.24 | -0.41 | 90.31 | -0.13 | 31.74 | 5.64 | 2.46 |

# References

38
[1]  Y. Chen, J. S. Emer, and V. Sze. Eyeriss v2: A flexible and high-performance accelerator for emerging deep neural
networks. *IEEE Journal on Emerging and Selected Topics in Circuits and Systems*, 2018.


[Meta-Review · NeurIPS 2019]

This paper proposes a distribution aware quantization which chooses between recentralized and shift quantizations based on weight distributions in the kernels. The proposed methods is novel, and provides a new general framework to quantize sparse CNNs. Experimental results are extensive and solid, and show the effectiveness of the proposed approach by comparing with the state-of-the-art on well known neural networks. There is also good ablation study. Moreover, the paper is well-written, except some figures are confusing.